# Species-Specific Sensitivity and Levels of Beta-D-Glucan for the Diagnosis of Candidemia—A Systematic Review and Meta-Analysis

**DOI:** 10.3390/jof11020149

**Published:** 2025-02-15

**Authors:** Nadir Ullah, Marco Muccio, Laura Magnasco, Chiara Sepulcri, Daniele Roberto Giacobbe, Antonio Vena, Matteo Bassetti, Malgorzata Mikulska

**Affiliations:** 1Department of Health Sciences (DISSAL), University of Genoa, 16132 Genoa, Italy; nadir.ullah@uvas.edu.pk (N.U.); marco112008@live.it (M.M.); chiara.sepulcri@gmail.com (C.S.); danieleroberto.giacobbe@unige.it (D.R.G.); anton.vena@gmail.com (A.V.); matteo.bassetti@hsanmartino.it (M.B.); 2UO Clinica Malattie Infettive, IRCCS Ospedale Policlinico San Martino, 16132 Genoa, Italy; laura.magnasco@hsanmartino.it

**Keywords:** (1,3)-ß-D-glucan, candidemia, sensitivity, *C. auris*, *C. albicans*

## Abstract

Background: 1, 3-ß-D-Glucan (BDG) is an antigen present in the cell wall of many pathogenic fungi and is used as a marker for the early diagnosis of candidemia and discontinuation of empirical treatment. Changes in the epidemiology of *Candida* species might have a negative impact on the performance of serum BDG. The aim of this study was to analyze the performance of BDG in candidemia diagnosis focusing on species-specific differences in BDG sensitivity and BDG levels. Methods: The PRISMA system was used for the systematic search. The following databases were searched for articles published from January 2010 to November 2023: PubMed, Science Direct, and Scopus. Results: A total of 21 studies that met the inclusion criteria were included, reporting data from 1633 patients with candidemia; 11 reported both sensitivity and specificity, 15 reported species-specific sensitivity, and nine reported species-specific BDG levels. The pooled sensitivity of BDG in all studies was 0.73 (95% confidence interval (CI), 0.66-0.80), while the pooled sensitivity and specificity in 11 studies were 0.81 (95% CI 0.73-0.89) and 0.80 (95% CI 0.74-0.87). BDG pooled sensitivity (all assays) and BDG levels (for assays with cutoff of 80 pg/mL) were the highest in *C. krusei* (currently *Pichia kudriavzevii*) and the lowest in *C. auris*: 0.76 and 417 pg/mL for *C. krusei*, 0.73 and 345 pg/mL for *C. albicans*, 0.74 and 356 pg/mL for *C. glabrata* (currently *Nakaseomyces glabrata*), 0.70 and 324 pg/mL for *C. tropicalis*, 0.63 and 95 pg/mL for *C. parapsilosis,* 0.51 and 62 pg/mL for *C. auris,* and 0.44 and 79 pg/mL for other *Candida* species. These differences were statistically significant for BDG sensitivity and levels of *C. albicans*, *C. glabrata*, *C. krusei*, and *C. tropicalis* compared to *C. auris*, *C. parapsilosis*, and other *Candida* species. Conclusion: The sensitivity of BDG in candidemia diagnosis depends on the *Candida* species, with the lowest being for *C. auris* and *C. parapsilosis*. This might have a clinical impact in centers where these species are prevalent.

## 1. Introduction

*Candida* species are unicellular fungi responsible for the most commonly encountered invasive fungal infections (IFI) among hospitalized patients admitted to various clinical settings in high-income countries, and candidemia is the most prevalent clinical presentation of invasive candidiasis (IC) [1,2,3]. The incidence of candidemia ranged between 1.2 and 25 cases per 100,000 persons in both European and American studies, with the incidence rate in Europe being lower [4,5,6]. A retrospective Italian study, together with a review of published data between 2010 and 2014, reported a statistically significant increase in the incidence of candidemia between 2008 and 2012 in hospital patients (0.4–1.68 per 10,000 patients–day) [7]. Candidemia remains a significant cause of mortality, with a mortality rate up to 40% in a recent multicenter observational cohort study [8]. To reduce the morbidity and mortality due to *Candida* infection, early diagnosis and prompt antifungal therapy are required [9,10].

The distribution of *Candida* species has changed over the last decade [11,12,13]. A meta-analysis by Giacobbe and colleagues [14] reported that the overall incidence of non-*albicans* species in all hospital-admitted patients was 49.5 (95% CI, 48.0–51), while the pooled incidence of non-*albicans* species in the ICU setting was 50.6% (95% CI 46.6–54.6). The shift in species epidemiology might have a significant negative impact on early diagnosis and management of IC [15]. Indeed, it might impair the performance of some non-culture-based diagnostic tools, e.g., molecular methods or T2 MRI, which were designed primarily for *C. albicans* and usually included only the main *Candida* species [16]. Additionally, changes in species prevalence are also fundamental for the appropriateness of empirical antifungal therapy, e.g., in the case of high rate of azole resistance in *C. parapsilosis*, or azole-resistant *C. auris*, with or without resistance to amphotericin B [11,17].

Blood cultures are still the gold standard for diagnosing candidemia; however, studies have reported variable and low sensitivity ranging from 27 to 71% [18]. 1,3-beta-D-glucan (BDG) is an early detection marker for candidemia that has been used for several years as a surrogate marker of IC [19,20]. BDG is a fungal polysaccharide found in the cell wall of many pathogenic fungi except for *Blastomyces dermatitidis*, *Zygomycetes,* and most *Cryptococci* [21]. The use of BDG as a marker of IFI such as candidemia or IC is recommended by international guidelines [22]. In a meta-analysis of studies published from 2009 to 2016 [23], BDG showed pooled sensitivity and specificity of, respectively, 0.81 (95% CI 0.73–0.87) and 0.61 (95% CI 0.46–0.75) for diagnosis of IC and candidemia in adult ICU-admitted patients. A systematic review and meta-analysis in neonatal patients with IC, including eight studies and 465 participants, reported a summary sensitivity and specificity of 0.89 (95% CI 0.80–0.94) and 0.60 (95% CI 0.54–0.66), respectively [24].

Several studies reported differences in BDG sensitivity for some *Candida* species, particularly for *C. parapsilosis* and *C. auris*. Recently, we reported BDG sensitivity for candidemia in patients admitted to the ICU and noted an important difference based on *Candida* species: 62% sensitivity for *C. albicans,* 42% for *C. auris,* and 49% for *C. parapsilosis* [25]. Similarly, another study by Farooqi and colleagues reported a significant difference between the sensitivity of BDG for *C. auris* (44%) and non-*auris Candida* species (e.g., 79% for *C. tropicalis* and 72% for *C. albicans*) [26]. Several studies reported lower BDG levels for *C. parapsilosis* compared to *C. albicans* [26,27,28].

Therefore, to address the issue of varying BDG sensitivity in various *Candida* species, we performed a meta-analysis to investigate the BDG performance in candidemia diagnosis and specifically focused on species-specific differences in BDG sensitivity and BDG levels.

## 2. Methods

This systematic review and meta-analysis followed the Preferred Reporting Items for Systematic Reviews and Meta-Analyses (PRISMA) guidelines [29].

### 2.1. Literature Selection Criteria

We conducted a systematic review of studies that reported data on serum BDG performance in the diagnosis of candidemia and reported data on serum BDG performance in different *Candida* species between the inception of the database and November 2023. The inclusion criteria included: I. Studies that reported BDG performance for diagnosis of candidemia; II. Studies that reported BDG sensitivity and BDG levels in various *Candida* species; III. Studies published in English language; IV. Studies with full-text available; V. If there was more than one study from a single center, they were evaluated for the overlap of included patients; in case of overlap, the study with a higher number of patients was included; VI. Studies that reported either only adults or the majority of the included population were adults; VII. Studies that reported sensitivity of any commercially available BDG assay were included if they provided results using the manufacturer cutoff value of the assay; VIII. All research articles published between 2015 to 2023; in addition, articles published between January 2010 and December 2014 were included if they reported species-specific differences in BDG sensitivity. The exclusion criteria were as follows: I. Studies not reporting BDG sensitivity; II. Studies that reported overall BDG performance for candidemia together with other invasive fungal diseases (IFD); III. Studies with a number of candidemia patients lower than 10; IV. Studies that focused only on neonatal or pediatric patients; V. Studies with results presented as abstract only.

### 2.2. Index Test and Reference Standard

Studies reporting commercially available BDG tests for candidemia diagnosis were included. For studies reporting test performance for multiple cutoff values, only data for the manufacturer-recommended cutoff value were extracted. If studies reported performance of more than one BDG assay, for meta-analysis, we extracted data for the main analyses only for one test, preferably the most frequently used one. Blood culture was the reference standard of choice for candidemia diagnosis. If, in addition to blood cultures, other criteria were considered as reference standard by a single study, they were also extracted but considered as potential sources of bias.

### 2.3. Search Strategy for Identification of the Studies

Three databases (PubMed, Science Direct, and Scopus) were searched by (UN and LM), using the following relevant keywords: “Candidemia” AND (diagnosis AND 1, 3 beta D Glucan AND BDG), and “1, 3 beta D glucan” (Sensitivity AND specificity AND *Candida* species) for the given systematic review and meta-analysis, and a filter was applied to narrow the search from January 2010 to November 2023 and limit the search to only English language articles. The results from all the e-databases after the filter were 124 and 108 for Science Direct, 35 and 14 for PubMed, and 161 and 103 for Scopus. We also cross-checked the references of previous meta-analyses and included papers. For all three databases, yearly distribution of published articles is presented in Appendix A.

### 2.4. Data Collection (Selection of Studies)

To collect the data, two review authors (UN and LM) screened the titles and abstracts of research papers to determine their suitability for inclusion. For the full-text review, we excluded articles considered irrelevant during the initial screening process but included those deemed relevant to the study aims. Differences between the two review authors were resolved by discussion. The information extracted included study design, number of candidemia patients, index test, sample for index testing, and reference method for diagnosing candidemia. All this information was reviewed, and any differences were resolved through discussion between the two review authors.

### 2.5. Data Extraction

Two review authors (UN and LM) extracted information from the selected papers on the number of patients included, the ward of admission or underlying condition (ICU, non-ICU, medical, surgical, cancer, mixed patient population, etc.), type of population (adults and adults/pediatric), design of the study (retrospective single/multicenter, prospective single/multicenter, and case-control study), BDG assay used (any FDA-approved or commercially available assay), sample type (serum only), fungal infection type (candidemia only), *Candida* species, cutoff used, and any other relevant additional information. All the details were recorded, and discrepancies were resolved through discussion with the main author (MM).

### 2.6. Methodological Quality Assessment

The Quality Assessment of Studies of Diagnostic Accuracy-Revised (QUADAS-2) tool was used to evaluate the study quality [30]. The bias in the included studies was assessed in four domains: patient selection, index test, reference standard, and flow/timing, while applicability concern was evaluated in the three domains: patient selections, index test, and reference standard. Two reviewers (UN and LM) independently rated bias and applicability concerns as low, high, or unclear. Discrepancies were then resolved through discussion with the main author (MM).

### 2.7. Statistical Analysis

Data was extracted, entered into a Microsoft Excel sheet, and analyzed using SAS software (version 9.4, SAS Institute Inc., Cary, NC, USA). BDG levels were reported as weighted medians along with weighted interquartile ranges. 2×2 matrices were constructed for sensitivity and specificity, with entries extrapolated from values reported in the reference article or calculated otherwise. Variances and confidence intervals for these measures were estimated using the Adjusted Wald method, and heterogeneity across studies was measured using the Q test and the estimation of I² statistics. Subsequently, formal meta-analyses were conducted, and either a fixed-effects or random-effects model, depending on the level of heterogeneity, was applied to calculate the pooled estimate for each sensitivity and specificity, their confidence interval, and the eventual value of variance between studies (τ²). The sources of heterogeneity for overall sensitivity and specificity were investigated using weighted fixed-effects models. Then, ROC plots were constructed where possible. Weighted fixed-effects models were also employed to analyze differences in BDG levels among different *Candida* species and to investigate potential sources of variation in total BDG levels across studies. A random-effect model was used to analyze differences in sensitivities among different *Candida* species. Forest plots were generated to represent the results visually. All statistical tests were two-sided, with a *p*-value ≤ 0.05 considered statistically significant.

## 3. Results

### 3.1. Results of the Search

As a result of the bibliographical search in PubMed, Scopus, and Science Direct, we identified 21 individual studies reporting the performance of BDG testing in candidemia diagnosis [25,26,27,28,31,32,33,34,35,36,37,38,39,40,41,42,43,44,45,46,47], that were included in our systematic review and meta-analysis. We contacted one study author [41] for clarification on the BDG assay used in the study, which we received. A flow chart diagram displaying the flow of the 21 included studies through the review is shown in Figure 1.

### 3.2. Characteristics of the Included Studies

Table 1 summarizes the main characteristics of the included studies [25,26,27,28,31,32,33,34,35,36,37,38,39,40,41,42,43,44,45,46,47]. One thousand six hundred thirty-three candidemia patients were included across these 21 studies, ranging per study from 10 to 218. One study reported both breakthrough candidemia and non-breakthrough candidemia; only non-breakthrough candidemia cases were included in the analysis [46]. The key characteristics of the studies and summary of the papers are presented in Table 1, and additional data is in Appendix A. Additional details of studies that reported mixed patient populations from different wards or did not specify the ward of admission are reported in Appendix A.

The majority of the studies were from Europe (n = 13): Italy (n = 4), Germany (n = 4), Spain (n = 3), France (n = 2) and the United Kingdom (n = 1). Two studies were reported from South Korea and one from each of the following: Brazil, Japan, Pakistan, South Africa, and Turkey. Twelve studies reported on mixed patient populations (57%), followed by three studies in ICU–admitted patients only (14%), three studies that did not specify the ward of admission (14%), and the remaining three reported patients from haematology–oncology (n = 1, 5%), internal medicine (n = 1, 5%), and any non-ICU (n = 1, 5%). Most studies included an adult population only (n = 17, 81%), while four included mixed adult and pediatric/neonatal population (n = 4, 19%). Of the 21 studies, more than half had retrospective single/multicenter design (n = 11, 52%). Eight of the remaining 10 studies had prospective single/multicenter design (n = 8, 38%), and two had case-control design (n = 2, 10%). More than half of the studies (n = 15, 71%) used Fungitell assay, followed by Wako (n = 3, 14%), Goldstream (n = 2, 10%), and Fungus (1-3)-β-D-Glucan Test (GCT-110T) (n = 1, 5%), (see Appendix A for the summary of the included studies).

### 3.3. Findings of the Study, Overall Pooled Sensitivity and Specificity, and Serum BDG Levels

The studies had high heterogeneity, and 11/21 provided both sensitivity and specificity. We included also studies that did not report the specificity because we aimed to document the differences in sensitivity across different *Candida* species. The pooled sensitivity in all included studies for candidemia diagnosis was 0.73 (95% CI 0.66–0.80), as shown in Table 2 and Figure 2, and the serum weighted median BDG concentration was 255 pg/mL (IQR 158.0–351.0) (reported for assays with cutoff of 80 pg/ml). In 11 studies that provided both sensitivity and specificity, the pooled sensitivity of BDG was 0.81 (95% CI 0.73–0.89), and pooled specificity was 0.80 (95% CI 0.74–0.87), Table 2 and Figure 3).

The overall sensitivity ranged from 41% to 100%, and specificity ranged from 39% to 98%. Fungitell assay had a sensitivity ranging from 44% to 100% and a specificity from 39% to 98%. For the Wako assay, the sensitivity ranged from 58% to 81%, and the specificity ranged from 85% to 93%. Two studies reported the sensitivity for Goldstream assay (41% and 87%) and one reported specificity of 55%, while one study reported sensitivity and specificity of Fungus (1-3)-β-D-Glucan Test (GCT-110T) of 82% and 55%, respectively.

In retrospective studies, the sensitivity of BDG assay ranged from 41% to 84%, and the specificity ranged from 55% to 93%. In the studies with prospective design, the sensitivity ranged from 63% to 100%, and the specificity ranged from 39% to 98%. In two case-control studies, the sensitivity was 87% and 91%, and the specificity was 85% and 88%.

In total, 3/21 studies specifically reported BDG performance in ICU patients. The sensitivity of the BDG ranged from 47% to 93%, and the specificity was 64%, while 12/21 studies reported data from patients admitted to different wards, with sensitivity between 41% and 100%, and the specificity between 39% and 98% (see Table 2 for the main results of the included studies). 

### 3.4. Methodological Quality Assessments of Included Studies

For 12 among 21 studies, there was no concern regarding the risk of bias and applicability. Four studies had unclear bias, mainly due to patient selection and index test. Five studies had a high risk of bias (in two studies related to applicability due to case-control design, and three studies related to flow and timing due to the exclusion of some patients from the final analysis). Overall, among all included studies, two studies failed to report a time frame between the reference standard and index test, while 19 studies reported a clear time frame between the reference standard and index test. In the case of the index test, we judged the studies as low risk of bias if they reported manufacture cutoff value for the index test, and if the index test for the majority of the patients was performed within 72h from the first positive blood culture. However, we reported an unclear risk of bias for the index test: if the index test was performed after 72h from the positive blood cultures. For the reference standard, low risk of bias was attributed if blood cultures were reported as reference method for diagnosis of candidemia. Details of the bias are included in Appendix A.

### 3.5. Investigations of Heterogeneity

Investigation of heterogeneity was evaluated by ROC plots that studied differences in individual sensitivity and specificity estimates by BDG assay type, ward of admission, and study design. ROC analysis was restricted to only 11 studies reporting sensitivity and specificity.

Heterogeneity did not differ based on the assay used (Appendix A). In seven studies using the Fungitell assay, sensitivity ranged from 72% to 100%, and specificity ranged from 39% to 98%. For two studies with the Wako test, sensitivity ranged from 58% to 81%, and specificity between 85% and 93%, with both assays reporting a wide range of sensitivity and specificity. For two other assays, the sensitivity and specificity were reported by a single study: for Goldstream the sensitivity and specificity were 87% and 75%, and forGCT-110T, sensitivity and specificity were 82% and 55%, respectively. Overall, similar sensitivity and specificity were reported for each assay.

In the analysis by the ward of admission in 11 studies, seven reported a mixed patient population, with sensitivity ranging from 58% to 100% and specificity from 39% to 98%. In one study including only ICU patients, the sensitivity was 93% and specificity was 64%. In two other studies including patients admitted to other wards, sensitivity and specificity were 91% and 88% in one with non-ICU patients, while sensitivity and specificity were 82% and 55% in one with internal medicine patients. The study that did not specify the ward of admission reported a sensitivity of 80% and a specificity of 75%. These results are shown in Appendix A.

Four retrospective studies reported sensitivity between 58% and 84%, and specificity between 55% and 93%; five prospective studies reported sensitivity between 72% and 100%, and specificity between 39% and 98%. Two case-control studies reported the following sensitivity and specificity: 87% and 85% for one study and 91% and 88% for another (Appendix A).

### 3.6. Distribution of Candida Species

Overall, 18 studies reported the distribution of *Candida* species, and the number of strains of each species is shown in Appendix A and Table 3. In 16/18 studies, *C. albicans* was the most frequent species, causing from 28% to 62% of all cases of candidemia. In two studies, *C. parapsilosis* was the most frequent species, responsible for 38% and 58% of candidemia episodes. The overall distribution of all species in 18 studies was as follows: *C. albicans,* 43%; *C. parapsilosis,* 23%; *Nakaseomyces glabrata* (formerly *C. glabrata*)*,* 13%; *C. tropicalis,* 9%; *C. auris,* 5%; *Pichia kudriavzevii* (formerly *C. krusei*)*,* 3%, and other *Candida* species, 3%.

### 3.7. Description of the Studies That Reported Species-Specific Sensitivity

A total of 15 studies provided BDG sensitivity values for each species separately. Table 3 shows the mean sensitivity of BDG for each *Candida* species, including the number of strains of different species included in the individual study and the BDG assay used. Sensitivity was reported for *C. albicans* in 15 studies, *C. parapsilosis* in 13, *N. glabrata* in 11, *C. tropicalis* in nine, *P. kudriavzevii* in eight, other *Candida* in seven, and for *C. auris* only in three studies.

### 3.8. Description of the Studies That Report Species-Specific Median BDG Levels

Overall, among 18 papers reporting the distribution of *Candida* species, nine (50%) reported BDG levels for different species: seven for the Fungitell assay only, one for the Goldstream assay, and one for both the Fungitell and Wako assays. Since Fungitell and Goldstream have the same cutoff for positivity of 80 pg/mL, species-specific median BDG levels were analyzed both together for these tests and separately for Fungitell. The species-specific BDG levels are presented in Appendix A, including the number of strains of different species included in each study and the BDG assay used. Additional details on other *Candida* species are presented in Appendix A. A total of nine studies reported *C. albicans* BDG levels, eight reported *C. parapsilosis* BDG levels, seven reported *N. glabrata* BDG levels, five reported *C. tropicalis* BDG levels, four reported *P. kudriavzevii* BDG levels, and only three studies reported *C. auris* and other *Candida* BDG levels.

### 3.9. Species-Specific Differences in BDG Sensitivity and BDG Blood Levels for All BDG Assays

We examined the pooled sensitivity and weighted median BDG levels across various *Candida* species. Table 4 indicates pooled sensitivity and weighted median BDG levels, including the number of *Candida* species isolates, range of sensitivity, and range of BDG levels. The overall pooled sensitivity and weighted median for each species are presented in Figure 4, Figure 5 and Figure 6, while the sensitivity for each single study is presented in Appendix A. The pooled sensitivity was 0.76 (95% CI 0.58–0.93) for *P. kudriavzevii*, 0.73 (95% CI 0.67–0.80) for *C. albicans*, 0.74 (95% CI 0.64–0.84) for *N. glabrata,* 0.70 (95% CI 0.52–0.88) for *C. tropicalis,* 0.63 (95% CI 0.52–0.73) for *C. parapsilosis*, 0.51 (95% CI 0.35–0.67) for *C. auris*, and 0.44 (95% CI 0.22–0.67) for other *Candida* species, respectively. The respective weighted median serum BDG concentrations were 417 pg/mL (IQR, 146–523) for *P. kudriavzevii,* 345 pg/mL (IQR 288–406) for *C. albicans,* 356 pg/mL (IQR 256–500) for *N. glabrata*, 324 pg/mL (IQR 324–846) for *C. tropicalis,* 95 pg/mL (IQR 78–407) for *C. parapsilosis,* 62 pg/mL (IQR 48–62) for *C. auris,* and 79 pg/mL (IQR 79–79) for other *Candida* species.

### 3.10. Species-Specific Differences in BDG Sensitivity and BDG Blood Levels for Fungitell Assay

The sub–group analysis was restricted to the Fungitell assay. The pooled sensitivity was 0.88 (95% CI, 0.76–1.00) for *P. kudriavzevii*, 0.83 (95% CI, 0.75–0.90) for *N. glabrata,* 0.80 (95% CI, 0.71–0.89) for *C. tropicalis*, 0.77 (95% CI, 0.71–0.84) for *C. albicans*, 0.61 (95% CI, 0.50–0.71) for *C. parapsilosis*, 0.51 (95% CI, 0.35–0.67) for *C. auris,* and 0.39 (95% CI, 0.05–0.74) for other *Candida* species, respectively. The weighted median serum BDG concentrations were 417 pg/mL (IQR, 146–523) for *P. kudriavzevii,* 345 pg/mL (IQR 288–406) for *C. albicans,* 356 pg/mL (IQR 256–500) for *N. glabrata*, 324 pg/mL (IQR 324–324) for *C. tropicalis,* 95 pg/mL (IQR 78–407) for *C. parapsilosis,* 62 pg/mL (IQR 48–62) for *C. auris,* and 79 pg/mL (IQR 79–79) for other *Candida* species. The pooled sensitivity is presented in Table 4 and Figure 5, and weighted median is presented in Table 4. In contrast, the sensitivity for each single study is presented in Appendix A.

### 3.11. Comparison of Different Candida Species

Specific-specific differences in the sensitivity between different *Candida* species were analyzed (one species versus one species comparison), and we found statistically significant differences between pooled sensitivities. When considering all BDG assays and the Fungitell assay alone, sensitivity for *C. albicans, P. kudriavzevii, N. glabrata,* and *C. tropicalis* candidemia exhibited significantly higher sensitivity compared to *C. auris, C. parapsilosis,* and other *Candida* species (*p* < 0.05 for all comparisons). No significant differences were observed among *C. albicans, P. kudriavzevii, N. glabrata,* and *C. tropicalis* candidemia in either assay (*p* > 0.05).

Similarly, we also found statistically significant differences in serum BDG levels in case of infection with different *Candida* species: higher levels in case of *C. albicans* or *P. kudriavzevii* or *N. glabrata* or *C. tropicalis* compared to *C. auris, C. parapsilosis,* and other *Candida* species (*p* < 0.05 for all comparisons). No statistically significant differences in BDG levels were observed among *C. albicans*, *P. kudriavzevii*, *N. glabrata,* and *C. tropicalis* (*p* > 0.05 for all 12 comparisons, which includes all BDG assays and Fungitell alone). The pooled sensitivity and weighted median BDG levels are presented in Table 4 and Figure 4, Figure 5 and Figure 6.

## 4. Discussion

### 4.1. Main Findings Part 1 (All Species)

The main finding of our systematic review and meta-analysis was the pooled sensitivity of BDG in candidemia diagnosis in all included studies was 0.73 (95% CI, 0.66–0.80), while the pooled sensitivity and pooled specificity of BDG for candidemia diagnosis among 11 studies that reported both sensitivity and specificity, the pooled BDG sensitivity was 0.81 (95% CI 0.73–0.89), and the pooled specificity was 0.80 (95% CI, 0.74–0.87).

#### Main Findings Part 2 (Different Species)

To the best of our knowledge, this is the first systematic review and meta-analysis to report differences in sensitivity and BDG levels in different *Candida* species. We found significant differences in the sensitivity and BDG levels for candidemia due to different *Candida* species, both when considering all BDG assays and the Fungitell assay alone. The species-specific pooled sensitivity was between 70% and 76% for *P. kudriavzevii*, *C. albicans, N. glabrata*, or *C. tropicalis,* while it was 63% for *C. parapsilosis,* only 51% for *C. auris,* and 44% for other *Candida* species. Also, serum BDG levels differed significantly, being in median over 300 pg/mL for *P. kudriavzevii* (417 pg/mL), *C. albicans* (345 pg/mL), *N. glabrata* (356 pg/mL), or *C. tropicalis* (324 pg/mL), compared to less than 100 for *C. parapsilosis* (95 pg/mL), *C. auris* (62 pg/mL), or other *Candida* species (79 pg/mL). Very similar results were obtained when the analyses were limited to Fungitell only, since it was the most frequently used assay.

The above findings should be interpreted in light of the high heterogeneity that we found among the studies. We analyzed the heterogeneity by examining group differences through ROC analysis focusing on the type of BDG assay, ward of admission, and study design, but we could not identify the specific cause of heterogeneity. One of the reasons might be that there were very few studies exclusively from one setting (e.g., ICU), and Fungitell was used in 15 of 21 studies.

The reasons for species-specific differences remain to be investigated. Interestingly, the species with lower BDG levels were *C. auris* and *C. parapsilosis,* which share some particular characteristic, such as colonized skin being the main source of blood entry, their propensity to cause nosocomial outbreaks, and higher minimal inhibitory concentration (MIC) values for echinocandins. In our experience, there were no data to suggest that lower serum BDG levels might be the result of certain candidemia cases being contamination of a central venous catheter, rather than true infection, as the BDG levels were still lower even if we considered only patients with blood cultures from peripheral vein-growing *Candida* [48]. Moreover, we found lower BDG levels also in in vitro tested strains of first, *C. parapsilosis* [49], and later also *C. auris* [50]. To better define the impact on clinical utility of serum BDG in the settings with predominance of *C. parapsilosis* or *C. auris* due to an outbreak, analyses of BDG sensitivity in case of different *C. auris* clades or *C. parapsilosis* strains might be useful. The association between higher MIC values for echinocandins, which inhibit beta-D-glucan synthetase [51], and the capacity of *Candida* strains to release BDG during infection has not been extensively studied. Further studies are needed to investigate this significant relationship and its consequence for the diagnosis and management of candidemia.

Recent development of molecular-level identification of fungi has led to the discovery that several clinically important species previously classified as *Candida* are now classified as other genera, such as C. *glabrata,* which is now *N. glabrata*, and *C. krusei,* which is now *P. kudriavzevii*, while *C. albicans*, *C. tropicali*s, *C. auris*, and *C. parapsilosis* are still members of the *Candida* genus according to the latest classification [52,53]. It is interesting that *C. auris* and *C. parapsilosis*, which are still in the *Candida* genus, have lower BDG levels than *C. tropicalis* and *C. albicans*.

Irrespective of the reasons for lower sensitivity in some species, these results are particularly important considering epidemiological changes with recent trends indicating an important rise in *C. parapsilosis* and *C. auris* prevalence. Indeed, we reported changes in species epidemiology between the 2008–2011 and 2012–2016 period. The number of candidemia episodes per 10,000 patient days increased significantly during the second observation period (from 1.97 to 4.59 episodes), primarily due to a rise in *C. parapsilosis* [11]. Similarly, a study from Kuwait reported a decreased incidence of *C. albicans* candidemia but an increase in *C. parapsilosis* prevalence [13]. Also a recent retrospective study from Greece reported an outbreak of *C. auris* candidemia in a hospital setting between 2021 and 2023, with *C. auris* being the most frequent cause of candidemia in 2023 (34%), affecting both i t ICU and non-ICU setting [12]. Similarly, another study from South Africa reported an increase in *C. auris* candidemia episodes [54]. Recently, in 2024, also our study group reported another change in distribution of *Candida* species, with an increase in *C. auris* infections between 2021 and 2023 [55].

These shifts in *Candida spp.* distribution may present new diagnostic challenges, since lower BDG sensitivity in case of *C. auris* and *C. parapsilosis* compared to *C. albicans*, *N. glabrata*, *P. kudriavzevii,* and *C. tropicalis* might lead to a higher rate of false-negative results of BDG testing.

### 4.2. Comparison with Previous Meta-Analysis

A meta-analysis from Haydour and colleagues published in 2019 [23] reported BDG pooled sensitivity in candidemia and IC patients admitted to ICU. Their main focus was to document the diagnostic accuracy of BDG in candidemia and IC in ICU patients, and they did not analyze species-specific differences in the BDG levels and sensitivity. They included 10 studies, and the pooled sensitivity and specificity were reported as 0.81 (95% CI 0.73–0.87), and 0.61 (95% CI 0.46–0.75), respectively. Despite no overlapping studies, the same overall sensitivity was found in our meta-analysis from the mixed patient population (0.81, 95% CI 0.73–0.89), while specificity was higher (0.80, 95% CI 0.74–0.87).

Another systematic review and meta-analysis by He and colleagues from 2015 [56] reported the diagnostic accuracy of BDG in all IFDs, including IC, but did not provide separate results for IC or candidemia patients. For all 896 patients with IFD, the pooled sensitivity and specificity were very similar to ours: 0.78 (95% CI 0.75–0.81) and 0.81 (95% CI 0.80–0.83).

Finally, another systematic review and meta-analysis by White and colleagues published in 2020 [57] reported the performance of BDG in immunocompromised or critically ill patients with IFD. They included 49 studies, of which 10 were specifically focused on *Candida* infections and included 1185 patients. The pooled sensitivity of serum BDG for IC and candidemia diagnosis was similar to ours - 0.81 (95% CI 0.75–0.86), while specificity was lower - 0.64 (95% CI 0.56–0.72) than in our meta-analysis. This might be explained by high rate (80%) of included studies being performed in the ICU setting, where different causes of false positivity can be present, such as dialysis, recent surgery, or immunoglobulin therapy. On the contrary, in our meta-analysis, only one study among those that reported specificity (9%) was performed exclusively in the ICU setting.

### 4.3. Limitations of the Study

Acknowledging the limitations of the present study is necessary for the interpretation of the findings accurately. First, the quality of the included studies was moderate and differed in many ways, which is commonly observed in retrospective studies on diagnostic techniques. Second, BDG accuracy for diagnosing candidemia may vary in different patient groups (ICU vs non-ICU, ICU vs onco-haematology, etc.), but we were unable to analyze these differences because most of the studies reported mixed patient populations. The third limitation of our study is that we only included full-text articles in the English language, which may have led to the exclusion of some papers reporting data from centers in which non-*albicans* species might be frequent (e.g., from Asia).

## 5. Conclusions

In conclusion, our systematic review and meta-analysis suggested that BDG has reasonable pooled sensitivity of 73% and specificity of 80% for diagnosing candidemia. However, this is the first meta-analysis that reported significant differences for different *Candida* species. *P. kudriavzevii, C. albicans, N. glabrata,* and *C. tropicalis* showed higher pooled sensitivity and BDG levels than *C. auris, C. parapsilosis,* and other *Candida* species. Such lower sensitivity of BDG for the diagnosis of candidemia caused by these species may have a negative clinical impact in centers where they are prevalent. Future research is needed to explore the reasons for these differences and their clinical importance in various clinical settings.

## Figures and Tables

**Figure 1 jof-11-00149-f001:**
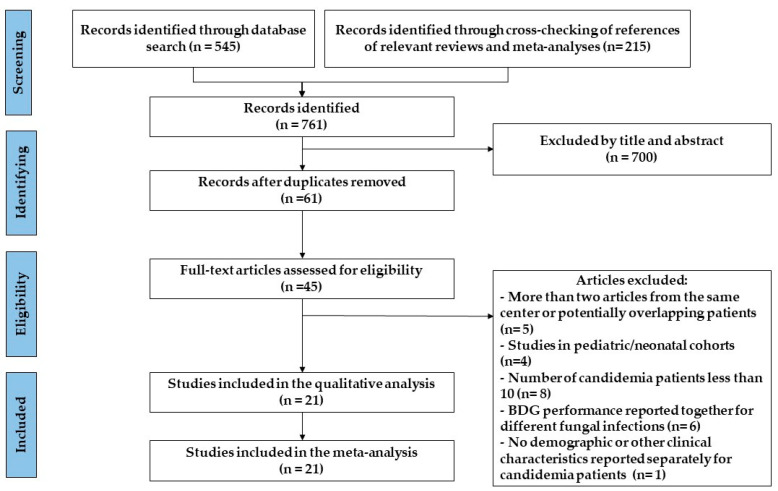
Flow chart of study selection.

**Figure 2 jof-11-00149-f002:**
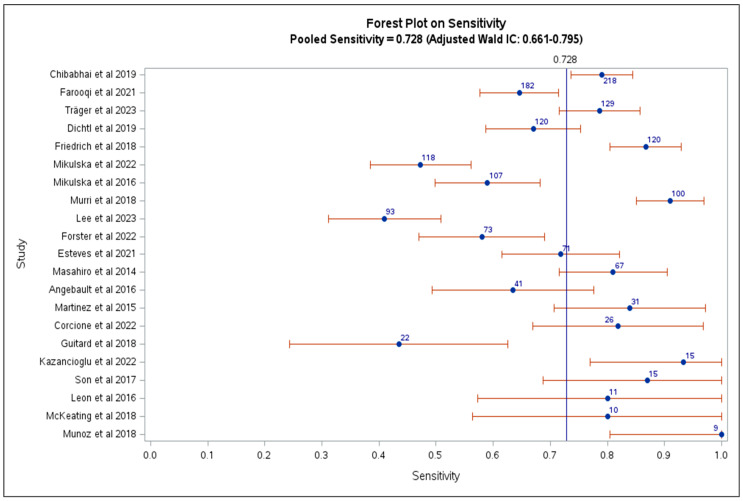
Pooled BDG sensitivity in the included studies, with the number of included patients reported in blue [25,26,27,28,31,32,33,34,35,36,37,38,39,40,41,42,43,44,45,46,47].

**Figure 3 jof-11-00149-f003:**
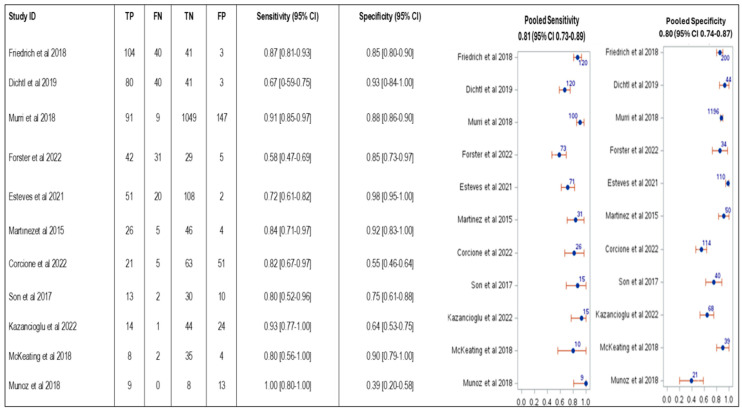
Pooled BDG sensitivity and specificity in 11 included studies, with the number of included patients reported in blue [28,33,34,35,36,37,39,40,41,43,47].

**Figure 4 jof-11-00149-f004:**
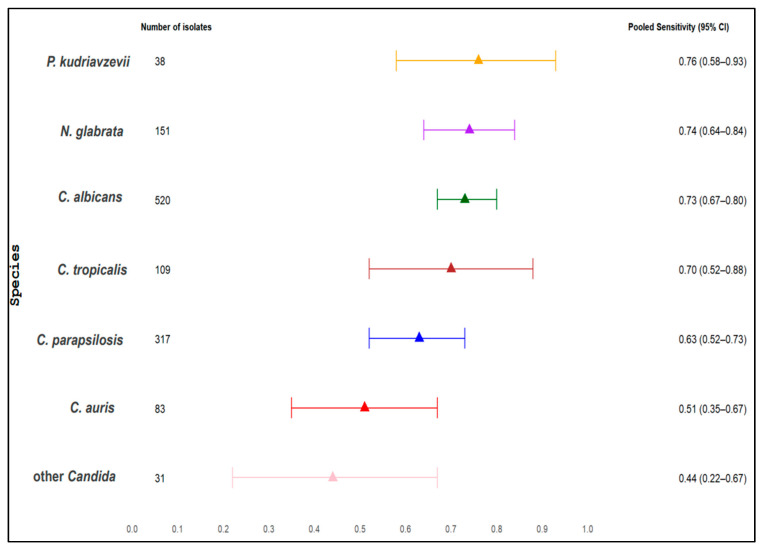
Pooled sensitivity of serum BDG for all *Candida* species and for all assays.

**Figure 5 jof-11-00149-f005:**
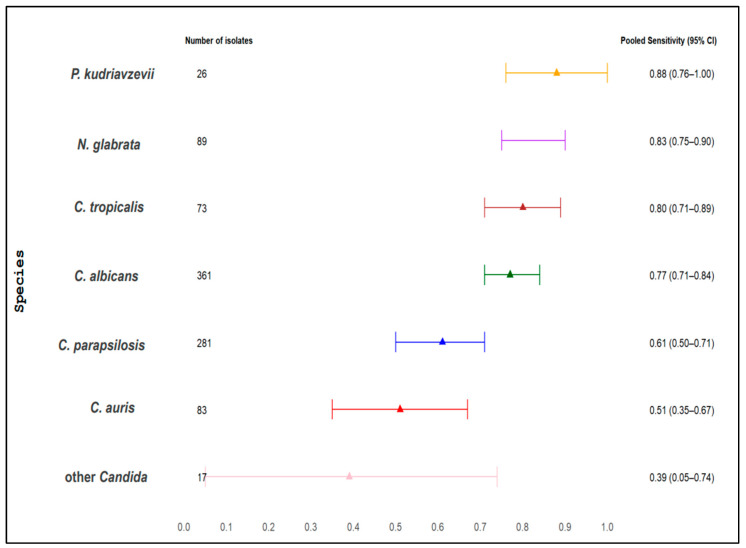
Pooled sensitivity of serum BDG for all *Candida* species and for the Fungitell assay.

**Figure 6 jof-11-00149-f006:**
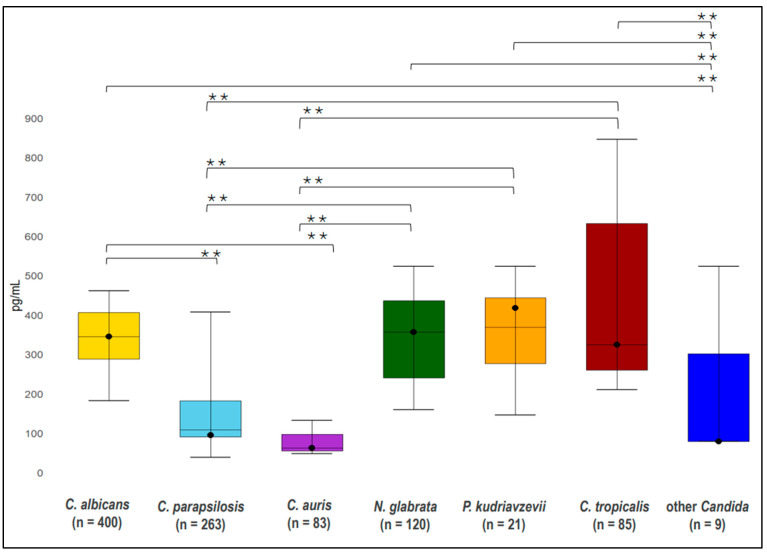
Weighted median BDG levels of different *Candida* species, reported for assays with cutoff of 80 pg/ml. The center line represents median BDG levels. The arrows above and below represent minimum and maximum median BDG levels. The black circle (●) represents the weighted median BDG levels. The star (*) represents the statistically significant difference between the different *Candida* species (*p* < 0.05).

**Table 1 jof-11-00149-t001:** Characteristics of the studies included in the meta-analysis, ordered according to the number of patients included.

Study ID	Study Design	Total Population Evaluated in the Study/ Included Patients and Control	Candidemia Only or Other	Candidemia Patients Met the Inclusion Criteria	Clinical Wards	Test and Cutoff for Positivity, pg/mL	Reported Both Sensitivity and Specificity	Reference Standard (Disease)
Chibabhai et al., 2019, South Africa [31]	Single-center retrospective	495/218	Candidemia only	218	Mixed (ICU = 137, Oncology = 21, Surgery = 17, not specified = 43)	Fungitell ≥ 80	Only sensitivity	Blood Culture (Candidemia)
Farooqi et al., 2021, Pakistan [26]	Single-center retrospective	505/182	Candidemia only	182	NA	Fungitell ≥ 80	Only sensitivity	Blood Culture (Candidemia)
Träger et al., 2023, Germany [32]	Prospective observational multicenter	138/129	Candidemia only	129	Mixed (ICU = 57, and for other ward of admission was not specified)	Fungitell ≥ 80	Only sensitivity	Blood culture (Candidemia)
Dichtl et al., 2019, Germany [33]	Single-center retrospective	120/120	Candidemia only	120	Mixed (ICU = 66, Transplant = 43, ward of admission was not specified = 11)	Wako ≥ 11	Both	Blood Culture (Candidemia)
Friedrich et al., 2018, Germany [34]	Case control single-center retrospective	383/383	Candidemia, bacteraemia, blood–culture negative group, and PJP	120	Mixed (ICU = 60, other not described = 60)	Fungitell ≥ 80	Both	Blood Culture (Candidemia)
Mikulska et al., 2022, Italy [25]	Single-center retrospective	255/118	Candidemia only	118	ICU	Fungitell ≥ 80	Only sensitivity	Blood Culture (Candidemia)
Mikulska et al., 2016, Italy [45]	Single-center retrospective	755/107	Candidemia only	107	Mixed (ICU/Surgery versus Medical/Haematology = 55, not specified = 52)	Fungitell ≥ 80	Only sensitivity	Blood Culture (Candidemia)
Murri et al., 2019, Italy [35]	Case control single-center retrospective	1296/1296	Candidemia and controls	100	Non-ICU	Fungitell ≥ 80	Both	Blood Culture (Candidemia)
Lee et al., 2023, South Korea [27]	Single-center retrospective	576/93	Candidemia only	93	Mixed (ICU = 42, Medical = 44, Surgery = 5, Emergency department = 2)	Goldstream ≥ 80	Only sensitivity	Blood Culture (Candidemia)
Forster et al., 2022, Germany [36]	Retrospective multicenter	82/82	Candidemia only	82	Mixed (ICU = 38, Non-ICU = 35, not specified = 9)	Wako ≥ 7*	Both	Blood Culture (Candidemia)
Masahiro Abe et al., 2014, Japan [46]	Single-center retrospective	147/92	Candidemia (breakthrough and non-breakthrough candidemia)	92	NA	Wako ≥ 11	Only sensitivity	Blood Culture (Candidemia)
Esteves et al., 2021, Brazil [37]	Prospective multicenter	241/181	Candidemia and healthy volunteers	71	Mixed (ICU = 30, Clinical = 18, Surgical = 9, not specified = 12)	Fungitell ≥ 80	Both	Blood Culture (Candidemia)
Angebault et al., 2016, France [38]	Prospective observational single-center	143/143	Candidemia, IC, IA and rare fungal IFDs	41	Mixed (ICU = 13, Haematology = 28)	Fungitell ≥ 80	Only sensitivity	Blood Culture (Candidemia) Computed tomography (IC)
Mart¡nez et al., 2015, Spain [39]	Single-center retrospective	81/81	Candidemia and bacteraemia	31	Mixed (ICU = 2, Surgical = 15, Medical = 10, onco haematology = 4)	Fungitell ≥ 80	Both	Blood Culture (Candidemia)
Munoz et al., 2018, Spain [40]	Prospective multicenter	44/30	Candidemia only	30	Mixed (ICU = 5, not specified = 25)	Fungitell ≥ 80	Both	Blood Culture (Candidemia)
Corcione et al., 2022, Italy [41]	Single-center observational retrospective and prospective	489/140	Candidemia and control group	26	Internal medicine	GCT-110T ≥ 80	Both	Blood Culture (Candidemia)
Guitard et al., 2018, France [42]	Single-center retrospective	41/34	Candidemia and CDC	22	Onco-haematological	Fungitell ≥ 80	Only sensitivity	Blood Culture (Candidemia) CT scan or ultrasonography (CDC)
Kazancioglu et al., 2022, Turkey [28]	Prospective single-center	137/83	Candidemia and non-candidemia	15	ICU	Fungitell ≥ 80	Both	Blood Culture (Candidemia)
Son et al., 2017, South Korea [43]	Prospective single-center	136/136	Candidemia, PJP, CDC, IA and mucormycosis	15	NA	Goldstream ≥ 80	Both	Blood Culture (Candidemia) Staining of liver biopsy or positive culture results from liver biopsy (IC)
Leon et al., 2016, Spain [44]	Prospective observational multicenter	322/233	Candidemia and candidiasis	11	ICU	Fungitell ≥ 80	Only sensitivity	Blood Culture (Candidemia) Macroscopic findings and direct examination or positive culture for *Candida* species of peritoneal fluid (IAC)
McKeating et al., 2018, UK [47]	Prospective Single-center	12/10	Candidemia	10	Mixed (ICU = 3, Surgical = 6, Oncology = 1)	Fungitell ≥ 80	Both	Blood Culture (Candidemia)

Abbreviations: CDC; chronic disseminated candidiasis: IA; invasive aspergillosis: IC; invasive candidiasis: ICU; intensive care unit: IAC; Intra-abdominal candidiasis: IFD; invasive fungal disease: NA; Not available: PJP; *Pneumocystis jirovecii* pneumonia. * Cutoff lowered as per manufactures instructions in 2022

**Table 2 jof-11-00149-t002:** The main results of the included studies.

	Sensitivity Range	Specificity Range	Pooled Sensitivity (95% CI)	Pooled Specificity (95% CI)	Description
Overall	41–100%	39–98%	0.73 (95% CI 0.66–0.80)	–	A total of 21 studies reported sensitivity.
Studies that reported both sensitivity and specificity	58–100%	39–98%	0.81 (0.73–0.89)	0.80 (0.74–0.8)	Of 21 studies, 11 reported both sensitivity and specificity.
Test Assays					
Fungitell, n = 15	44–100%	39–98%	–	–	Of 15 studies, seven reported both sensitivity and specificity.
Wako, n = 3	58–81%	85 and 93%	–	–	Two studies reported both sensitivity and specificity.
Goldstream, n = 2	41 and 87%	55%	–	–	One reported both sensitivity and specificity.
GCT-110T, n = 1	82%	55%	–	–	Sensitivity and specificity were reported
Study Design					
Retrospective, n = 11	41%–84%	55%–93%	–	–	Four reported sensitivity and specificity.
Prospective, n = 8	63%–100%	39%–98%	–	–	Five reported sensitivity and specificity.
Case-control study, n = 2	87 and 91%	85 and 88%	–	–	Both reported sensitivity and specificity.
Ward of admission					
ICU, n = 3	47%–93%	64%	–	–	One of three studies reported both sensitivity and specificity.
Mixed, n = 12	41%–100%	39%–98%	–	–	Seven studies reported both sensitivity and specificity.
Other wards, n = 3	44%–100%	55%–88%	–	–	Three studies reported patients from non-ICU, internal medicine, and haematology–oncology wards; two reported sensitivity and specificity.
Not specified, n = 3	65%–87%	75%	–	–	One reported sensitivity and specificity.

Abbreviations: ICU; intensive care unit

**Table 3 jof-11-00149-t003:** Species-specific sensitivity of (1, 3)-ß-D-Glucan in candidemia patients, with data on BDG assay type from 18 studies.

Study ID	Species-Specific Sensitivity (Number Inside the Brackets Shows the Number of *Candida* Isolates for Each Species)							
	BDG Assay Type	*C. albicans*	*C. parapsilosis*	*C. auris*	*N. glabrata* *(C. glabrata)*	*P. kudriavzevii* *(C. krusei)*	*C. tropicalis*	Other *Candida* Species
Träger et al., 2023 [32]	Fungitell	83% (70)	64% (11)	—	81% (31)	67% (6)	83% (6)	33% (3)
Lee et al., 2023 [27]	Goldstream	57% (28)	29% (7)	—	43% (21)	33% (6)	28% (25)	33% (6)
Mikulska et al., 2022 [25]	Fungitell	62% (29)	44% (84)	43% (21)	CS% (7)	—	CS (4)	CS (1)
Forster et al., 2022 [36]	Wako	57% (47) *	CS (7)	—	CS% (8)	CS (5)	CS (4)	CS (3)
Kazancioglu et al., 2022 [28]	Fungitell	100% (5)	80% (5)	—	100% (1)	100% (3)	—	100% (1)
Farooqi et al., 2021 [26]	Fungitell	72% (54)	53% (32)	44% (48)	88% (8)	—	79% (43)	—
Esteves et al., 2021 [37]	Fungitell	81% (36)	67% (15)	—	67% (6)	67% (3)	63% (8)	— (2)
Chibabhai et al., 2019 [31]	Fungitell	81% (81)	72% (83)	71% (14)	90% (30)	100% (10)	—	—
Dichtl et al., 2019 [33]	Wako	64% (58)	73% (11)	—	67% (27)	71% (7)	78% (9)	63% (4)
Friedrich et al., 2018 [34]	Fungitell	— (71)	— (10)	—	— (25)	— (2)	— (8)	— (7)
Friedrich et al., 2018 [34]	Wako	— (71)	— (10)	—	— (25)	— (2)	— (8)	— (7)
Murri et al., 2018 [35]	Fungitell	— (62)	— (24)	—	— (5)	—	—	—
Munoz et al., 2018 [40]	Fungitell	— (13)	— (9)	—	— (5)	— (1)	— (2)	—
Guitard et al., 2018 [42]	Fungitell	43% (7)	50% (4)	—	—	—	50% (6)	33% (4) **
McKeating et al., 2018 [47]	Fungitell	80% (5 ***)	— (1) +	—	75% (4)	—	—	—
Mikulska et al., 2016 [45]	Fungitell	72% (46)	41% (37)	—	— (10)	— (4)	— (7)	— (3)
Angebault et al., 2016 [38]	Fungitell	94% (16)	75% (4)	—	50% (4)	67% (3)	100% (3)	11% (9)
Martínez et al., 2015 [39]	Fungitell	67% (12)	83% (6)	—	75% (5)	100% (1)	100% (7)	—
Masahiro et al., 2014 [46]	Wako	81% (26)	89% (18)		79% (14)	—	50% (2)	75% (4)

Abbreviations: BDG: (1,3)-ß-D-Glucan; CS: combined sensitivity with other species; —: not reported: *; shows BDG sensitivity for 47 of 54 isolates: ^**^; Shows BDG sensitivity for four isolates of six: ***; shows BDG sensitivity for five isolates of six; +: shows *C. albicans* and *C. parapsilosis* (co-infections)

**Table 4 jof-11-00149-t004:** Pooled sensitivity, weighted median, and weighted mean of median BDG levels across different *Candida* species.

Candida Species	No. of Candida Strains for Which Pooled Sensitivity Was Reported	Range of Sensitivity Reported by Single Studies	Pooled Sensitivity (95% CI)	No. of Candida Strains for Which Weighted Median BDG Levels Were Reported	Range of Median BDG Levels Reported by Single Studies, pg/mL	Weighted Median of Median BDG and IQR, pg/mL
**All BDG assays**						
*C. albicans*	520	43%–100%	0.73 (0.67–0.80)	400	182–461	345 (288–406)
*C. parapsilosis*	317	29%–89%	0.63 (0.52–0.73)	263	39–407	95 (78–407)
*C. auris*	83	44%–71%	0.51 (0.35–0.67)	83	48–132	62 (48–62)
*N. glabrata*	151	43%–100%	0.74 (0.64–0.84)	120	159–523	356 (256–500)
*P. kudriavzevii*	38	33%–100%	0.76 (0.58–0.93)	21	146–523	417 (146–523)
*C. tropicalis*	109	28%–100%	0.70 (0.52–0.88)	85	210–846	324 (324–846)
Other *Candida*	31	11%–100%	0.44 (0.22–0.67)	9	79–523	79 (79–79)
**Fungitell assay**						
*C. albicans*	361	43%–100%	0.77 (0.71–0.84)	372	182–461	345 (288–406)
*C. parapsilosis*	281	41%–83%	0.61 (0.50–0.71)	256	39–407	95 (78–407)
*C. auris*	83	44%–71%	0.51 (0.35–0.67)	83	48–132	62 (48–62)
*N. glabrata*	89	50%–100%	0.83 (0.75–0.90)	99	224–523	356 (256–500)
*P. kudriavzevii*	26	67%–100%	0.88 (0.76–1.00)	21	146–523	417 (146–523)
*C. tropicalis*	73	50%–100%	0.80 (0.71–0.89)	60	210–632	324 (324–324)
Other *Candida*	17	11%–100%	0.39 (0.05–0.74)	9	79–523	79 (79–79)

**Abbreviations**: CI; confidence Interval: IQR; interquartile range.

## Data Availability

Upon reasonable request, data can be shared by contacting the corresponding author.

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
