# Peer review of "Species-Specific Sensitivity and Levels of Beta-D-Glucan for the Diagnosis of Candidemia—A Systematic Review and Meta-Analysis"

_jof, 2025, doi:10.3390/jof11020149_

Round 1
Reviewer 1 Report
The authors have presented a very important study evaluating the role of BDG in detecting Candida species. This is the first systematic review and meta-analysis to report species-specific differences in sensitivity and BDG levels across various Candida species. These findings highlight the importance of considering these variations in clinical practice to improve diagnosis and management.
1- Introduction:
The introduction is well-written but could be more concise. Many of the references used here could instead be included in the discussion section to strengthen it.
Line 64:
What is the relevance of mentioning colonization in this context?
2- The text demonstrates an adequate technical level and functional English; however, there are areas that could be further improved. Repetition of terms like "BDG sensitivity" and "species-specific" (lines 97, 315, 372) detracts from the flow of the text. Consider combining or rephrasing such phrases for better conciseness.
3- I suggest reviewing the redundancy of information between the two paragraphs in lines 369 to 381 to ensure clarity and avoid repetition.
4- Table S4:
The table titled “Species-specific sensitivity of (1,3)-ß-D-Glucan in candidemia patients, with data on BDG assay type from 18 studies” should be included in the main manuscript rather than the supplementary materials.
5- Lines 435–438:
Please include a reference to support the statement in these lines.
6- Line 456:
Why does the study by White, S.K., et al., 2020 report a specificity of 0.64? Please clarify or expand on this point.
7- Conclusion:
I have some doubts regarding the statement: “In conclusion, our systematic review and meta-analysis suggested that BDG has good pooled sensitivity and specificity for diagnosing candidemia.”
What criterion did you use to define the sensitivity and specificity as "good"?
Author Response
The authors have presented a very important study evaluating the role of BDG in detecting Candida species. This is the first systematic review and meta-analysis to report species-specific differences in sensitivity and BDG levels across various Candida species. These findings highlight the importance of considering these variations in clinical practice to improve diagnosis and management.
Comment 1: [The introduction is well-written but could be more concise. Many of the references used here could instead be included in the discussion section to strengthen it].
Response 1: [Thank you for the comment. We have moved some of the references from introduction to discussion section]. Please refer to the highlighted text in discussion part. (see line 433 to 449).
Comment 2: [Line 64: What is the relevance of mentioning colonization in this context?].
Response 2: [The authors are thankful for the reviewer comment]. The study cited included both colonized and infected patients. We reported in the context of changing species epidemiology. We rephrased the sentence to make it clearer. (Line 444-445).
Comment 3: [The text demonstrates an adequate technical level and functional English; however, there are areas that could be further improved. Repetition of terms like "BDG sensitivity" and "species-specific" (lines 97, 315, 372) detracts from the flow of the text. Consider combining or rephrasing such phrases for better conciseness].
Response 3: [The authors are thankful to the reviewer.Line 97: Finally, other studies, such as studies from Farooqi and colleagues, Lee and colleagues, and Kazancioglu and colleagues, reported lower BDG levels for C. parapsilosis compared to C. albicans. Line 315: While 18 studies reported the distribution of Candida species, only 15 reported BDG sensitivity values separately for different species. Mean species-specific BDG sensitivity is shown in Table S4]. Please refer to the highlighted text. We rephrase the statement, as per reviewer suggestion. Line 78-79: Finally, several studies reported lower BDG levels for C. parapsilosis compared to C. albicans. Line 295-296: A total of 15 studies provided BDG sensitivity values for each species separately. Table 3 shows the mean sensitivity of BDG for each Candida species. Line 372: Complete paragraph is rephrase from line 354 to 369.
Comment 4: [I suggest reviewing the redundancy of information between the two paragraphs in lines 369 to 381 to ensure clarity and avoid repetition].
Response 4: [The authors are thankful to the reviewer suggestion. The reviewer is right, the two paragraphs were bit confusing. We re-write the two paragraphs, as per reviewer suggestion]. See the highlighted text. (Line 354 to 369).
Comment 5: [The table titled “Species-specific sensitivity of (1,3)-ß-D-Glucan in candidemia patients, with data on BDG assay type from 18 studies” should be included in the main manuscript rather than the supplementary materials].
Response 5: [Thank you very much for the comment. We added the S4 table into the main text]. See the main revised manuscript. The table number is now 3, in the main manuscript.
Comment 6: [Lines 435–438: Please include a reference to support the statement in these lines].
Response 6: [The authors are thankful for the review comment. Reference has been added]. Reference is added. See the highlighted reference.
Comment 7: [Line 456: Why does the study by White, S.K., et al., 2020 report a specificity of 0.64? Please clarify or expand on this point].
Response 7: [The author are thankful to the reviewer comment]. We have expand on this point further. Please see the highlighted text. (Line 469 to 474).
Comment 8: [I have some doubts regarding the statement: “In conclusion, our systematic review and meta-analysis suggested that BDG has good pooled sensitivity and specificity for diagnosing candidemia.”
What criterion did you use to define the sensitivity and specificity as "good"?].
Response 8: [The author are thankful to the reviewer comment. We have updated the term in the revised manuscript]. We have changed the term Good into reasonable pooled sensitivity and pooled specificity studies. Usually the sensitivity and specificity >75 is considered reasonable. (Line 486-487).
Reviewer 2 Report
Dear authors,
Your work is current, excellently written and conducted, and has a very well-described methodology. Invasive candidiasis is a significant problem for diagnosis and treatment, and BDG, as an early detection marker for candidemia, is important for diagnosing invasive fungal infections. In this paper, you provide very useful information for this field of research.
In the following part, further comments for improving your manuscript are provided:
-In line 55, albicans should be written in italics
- In line 67, C. auris should be written in italics
- In line 94, auris should be written in italics
- in line 239 “the” should be deleted
- In Figure 6 all Candida species should be written in italics
- With the recent adoption of molecular identification of fungi, it has been established that several medically important species that were previously members of the genus Candida now belong to other genera (examples: Candida glabrata is now Nakaseomyces glabrata, Candida krusei - Pichia kudriavzevii, Candida lusitaniae - Clavispora lusitaniae, etc.). Even though many authors believe that old terms should still be used because the new taxonomy could confuse clinicians in interpreting the results of mycological analyses, a new classification needs to be introduced. It is very interesting that C. auris and C. parapsilosis, which are still members of the Candida genus according to the latest classification as non-albicans species, show lower sensitivity of BGD levels, and that can also be pointed out.

Author Response
Your work is current, excellently written and conducted, and has a very well-described methodology. Invasive candidiasis is a significant problem for diagnosis and treatment, and BDG, as an early detection marker for candidemia, is important for diagnosing invasive fungal infections. In this paper, you provide very useful information for this field of research.
Comment 1: [In the following part, further comments for improving your manuscript are provided: -In line 55, albicans should be written in italics - In line 67, C. auris should be written in italics - In line 94, auris should be written in italics - in line 239 “the” should be deleted - In Figure 6 all Candida species should be written in italics].
Response 1: [The authors are thankful for the reviewer comments and suggestions. The changes have been made in the revised manuscript]. Please refer to the highlighted text. All the term as mentioned by the reviewer changed into italic (Line 49 and line 77). The word (The) is deleted (Line 220) and highlighted in yellow. In Figure 6 the species names changed into italic. Please see the Figure 6 in revised manuscript.
Comment 2: [With the recent adoption of molecular identification of fungi, it has been established that several medically important species that were previously members of the genus Candida now belong to other genera (examples: Candida glabrata is now Nakaseomyces glabrata, Candida krusei - Pichia kudriavzevii, Candida lusitaniae - Clavispora lusitaniae, etc.). Even though many authors believe that old terms should still be used because the new taxonomy could confuse clinicians in interpreting the results of mycological analyses, a new classification needs to be introduced. It is very interesting that C. auris and C. parapsilosis, which are still members of the Candida genus according to the latest classification as non-albicans species, show lower sensitivity of BGD levels, and that can also be pointed out].
Response 2: [The authors are thankful for the reviewer comments and suggestions. The changes have been made in the revised manuscript]. Please refer to the highlighted text. Species name were updated throughout the main manuscript and in supplementary file. The name of the species was also updated in the Figures and Tables as well, both in main manuscript and supplementary file. Also the point is highlighted in the discussion section as well.